# Fintech, Digitalization, and Blockchain in Islamic Finance: Retrospective Investigation

Ibrahim Musa Unal and Ahmet Faruk Aysan *

Islamic Finance and Economy Program, Qatar Foundation, College of Islamic Studies, Hamad Bin Khalifa University, Ar-Rayyan 34110, Qatar
* Correspondence: aaysan@hbku.edu.qa

**Abstract:** The increasing interest in Fintech, Blockchain, and Digitalization in Islamic Finance created a new area in the literature, requiring a systematic review of these academic publications. The scope of the analysis is limited to journal articles to understand the trends in the indexed journals. Results are categorized into three sections, Islamic banks' digitalization, Blockchain and Crypto Assets research, and Islamic non-bank financial institutions' digitalization. Islamic fintech has great potential mainly because of the overlapping norms of Shariah and fintech, making it easier to implement technological disruption into Islamic finance. Moreover, the trust shift to Islamic finance could be merged with the opportunities of fintech and increase the potential of Islamic fintech even more.

**Keywords:** fintech; blockchain; digitalization; Islamic finance; shariah; Islamic banking; digital transformation; banking

## 1. Introduction

Although finance and banking have existed for centuries, technology has given them a new face with fintech and digitalization. Since the advent of ATMs and credit cards, banking services have been transformed in many steps and have absorbed the technology at its core business, also known as the Bank 4.0 concept [1]. Nowadays, digitalization has become necessary for all banks to keep pace, not only with the competition within the sector concerning other banks but with tech giants, start-ups, fintechs, and media-telecommunication companies.

The definition of a digitalized bank starts referring to increasing competition related to digitalization and joining more types of businesses in the game [2]. In the past, a bank with more credit card services and ATM locations would be accepted as "more digitalized." However, nowadays, financial institutions need high machine-learning skills to deal with the massive amount of big data, autonomous financial systems, and customer care where AI is actively used, and open-minded business strategies where innovations cooperate with blockchain and DLT-based technologies. Moreover, banks must do all these and many more while maintaining a smooth customer experience.

With digitalization, many other business fields established a financial system within their institution, such as tech giants, media-telecommunication companies, fintechs, and many different types of start-ups. This led to a significant change in the finance competition's rules, forcing the over-200-years-old finance industry to adapt to a new ecosystem.

Certainly, restructuring a mature industry is a painful process. Widharto et al. [3] explain a bank's scope of digital transformation in the following steps: adopting digital technology, redesigning processes for downsizing service costs, dominating the financial aspects with digital services, and restructuring corporate bodies. These steps show that a proper digital transformation for a bank/financial institution requires a top-to-bottom change in the organizational structure and business models. One of the main challenges in this transformation is completing this transformation while keeping the bank profitable

in a sector where finance is one of the most competitive and regulated industries. While start-ups and fintechs are flexible enough to make these changes due to their smaller size and agile structure, large and old institutions face much more challenges in adapting to the digitalized finance environment. As a result of this fact, even micro-scale and newborn fintechs could be a severe threat to billion-dollar banks, depending on their business models [4].

The size advantage also differs among small and large banks. A smaller bank will be more able to change the operational structure [5]. Since Islamic finance is relatively much younger and smaller in size than its conventional counterpart, opportunities for Islamic banks are more in terms of catching success through a digital transformation which is also agreed with an academic consensus in the literature [6].

Overall, the new concepts and technologies coming with the Banking 4.0 are clear game changers, and institutions are well aware of this. A clear sign of this awareness is seen in the recent surge in academic studies in the financial digitalization and fintech areas. Figure 1 shows the number of academic publications reviewed in this systematic literature review, illustrating the growth in the number of papers.

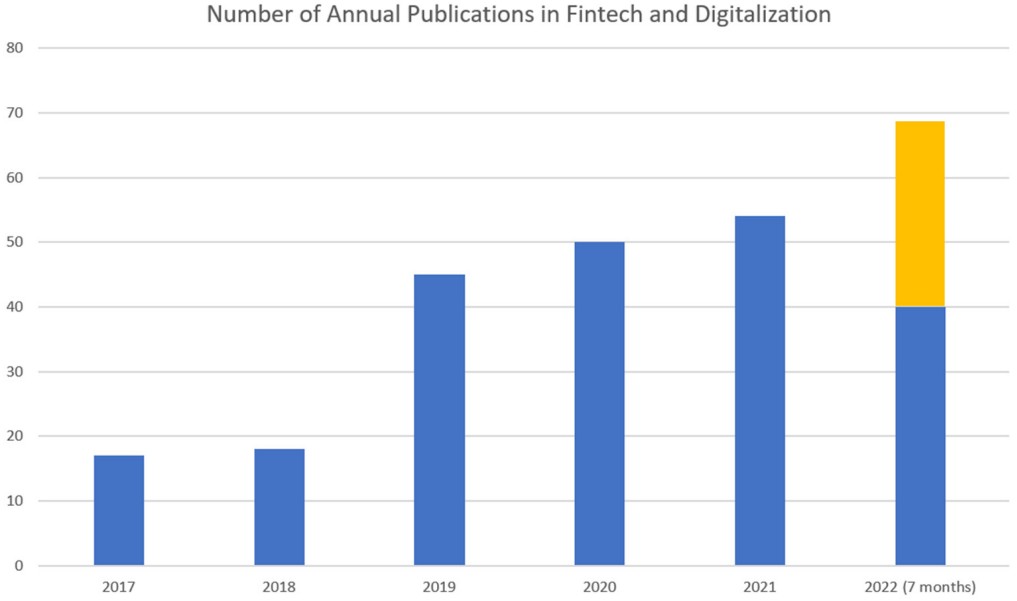

**Figure 1.** The number of Academic publications in the Islamic Fintech and Digitalization area (Yellow indicator is for a 12-month projection). Source: Authors' Calculations.

This increasing interest created a new area in the literature, requiring a systematic review of these academic publications, which is the main aim of this paper. According to the Vantage market research conducted in 2019, the global fintech market is expected to reach $332.5B by 2028. However, in such a booming industry, there is still a lack of international consensus over the term Fintech [7]. This confusion puts banks and researchers in a difficult position regarding the future direction of investments and research. Institutions dragging their feet in investing in fintech causes insufficient technology to meet the tremendous demand [8]. For fintech to be a blessing for these institutions, required investment should be sustained, and directed research should exist.

In fast-growing research areas and industries, there is often a lack of appropriate categorization and summarization of the current, which makes it difficult for the researchers to see a clear future strategy. In this regard, this paper aims to complete a comprehensive review of the existing literature on Islamic fintech and categorize the current work to give a clear snapshot of the trends. From this perspective, it is one of the first attempts in the Islamic finance literature and is aimed to be a lighthouse for researchers in their further studies.

## 2. Data and Methodology

The article focuses on the Scopus and Google databases with Islamic financial institutions for this goal, whereas fintech, blockchain, and digitalization titles are analyzed. Publications are limited from 2017 to 2022 to keep this review current. With the initial search, over 200 articles were found with the credentials. However, this number has been reduced to 100 after a manual election of the articles. In addition, the scope of the analysis is limited to journal articles to understand the trends in indexed journals.

Articles are categorized into three main groups. Figure 2 shows the areas covered in categories and the number of publications in each category. Category A focuses on Islamic banks' digitalization and fintech adaptation research. The banks are separated into different categories because of the high volume of this research and some technical differences between conducting bank research and non-bank institution research. Category B focuses on the research based on Blockchain, DLT, and digital currency research. Category C covers the digitalization of Islamic non-bank financial institutions and fintech articles, as well as Islamic micro/macro-economic related digitalization.

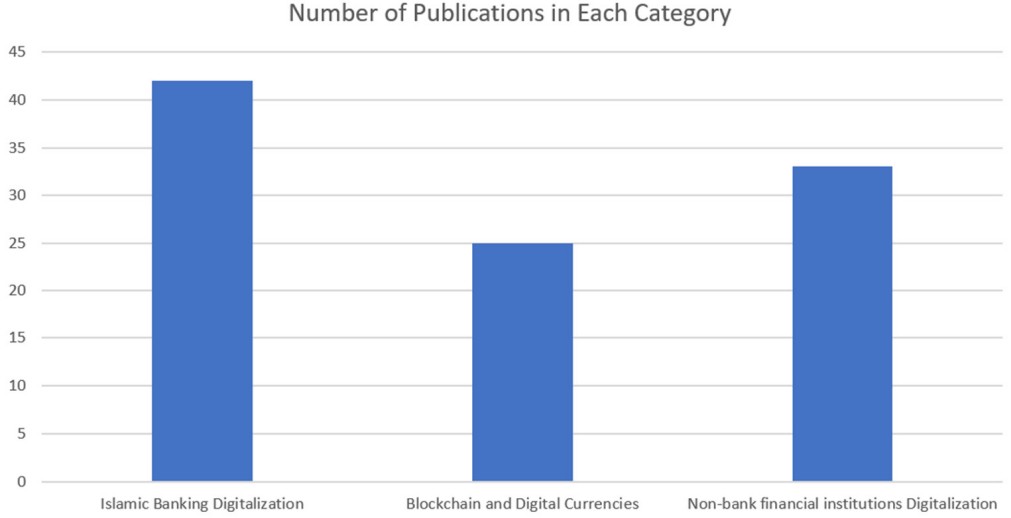

**Figure 2.** Number of Publications in Categories. Source: Authors' Calculations.

Categorization is carefully completed by reviewing each article and dividing them by the sub-industries of Islamic finance by relevance. These categories are selected by detecting the main focus points of Islamic fintech research. Most of the research is under these three categories, and the sub-industries of Islamic finance choose these categories. For example, all banking-related fintech articles are fitted in category A. Category B covers an area where all technologies are blockchain or decentralized philosophy dominates. Lastly, category C covers all non-bank institution-related fintech articles such as Islamic microfinance, Islamic SMEs, or crowdfunding. Smaller categories by research, such as risk management or Islamic circular economy, are included in the selected main categories, as the study about fintech and these categories are insufficient for creating a new category.

## 3. Literature Review

a. Fintech Adoption and Digitalization in Islamic Banking

Doubtlessly, a discussion about Islamic finance and digitalization must include Islamic Fintech. Islamic finance is a younger and more dynamic ecosystem than conventional finance. This fact opens the door for higher chances of innovation in Islamic Fintech, as justified above in this paper [7,9,10]. Majority of the literature that discusses Islamic Fintech focuses on its opportunities and potential [11–13]. However, many papers also focus on the challenges and threats [9,14]. The sufficient number of studies on both sides helps researchers see both the opportunities and challenges in Islamic Fintech and better focus on the necessary research.

Islamic Fintech literature is large enough to separate into two parts, Fintech for Islamic banks and non-bank institutions or countries. This section of the paper will cover Fintech for Islamic banks only, as it requires particular attention. Banking has experienced rapid change, and technology has become a survival must for all banks in the last two to three decades. Certain technologies are at the heart of this transformation, including AI, machine learning, big data, mobile internet, blockchain, and other DLT technologies. Aysan et al. [10] explained in their paper that most Islamic banks are performing stagnant at utilizing these technologies, except for mobile banking. The main reason for this technicality is that the mobile banking infrastructure is widely used; however, other technologies mentioned here require significant R&D and infrastructure investment. Indeed, these investments will be low in ROI as they are in the early stage [4].

In contrast, banks are not eager to transfer significant resources in a competitive and fragile environment [15]. Therefore, recent research shows that Islamic banks prefer to cooperate with young fintech and start-ups instead of investing directly in fintech [3]. This is because young fintechs and start-ups are safer to cooperate with or acquire if they prove their success. This increases the potential ROI for fintech implementation for Islamic banks and creates an encouraging environment for Islamic fintechs and start-ups.

On the flip side of the coin, Islamic Fintech has a much higher adaptation rate than conventional finance, as fintech and digital transformation are also supported by the principal doctrines of Islam, Maqasid al-Shariah [16]. Islam calls for the protection of the wealth of Muslims around the globe, which requires lower costs and friction in banking transactions. Fintech utilization is at the heart of reducing banking costs; therefore, Islamic banks are encouraged to use fintech in operations.

Islamic fintech is also mentioned as more transparent by its norms [17,18]. Islamic finance has gained a significant growth speed after the subprime crisis, related to losing trust in the conventional system. Shariah is very strict about ethical business and transparency; therefore, users started to shift to Islamic finance, especially in the last decade, which is caused by the belief that Islamic finance will make the financial world a more trustable environment [19]. This confidence brings enormous potential to Islamic fintech, possibly over 150 million new customers in the next three years [20].

The growth of Islamic fintech and banking customers is not only caused by the increasing trust of existing users but also by the booming Muslim population around the globe. Muslims are projected to be over 3 billion by 2060, and most Muslim countries have significant growth rates [21]. Hence, Islamic financial assets are expected to grow at high speed. However, this positive attitude towards Islamic fintech comes with question marks. Such high growth in a young industry requires immense human resources, a regulatory base, and a clear vision by governments [22]. Therefore, governments and academic institutions have an essential role in taking Islamic fintech to a successful future.

Besides the lack of trained personnel, there is undoubtedly a lack of authentic research in Islamic Fintech [23]. Although, as mentioned at the beginning of the paper, academic research institutions and universities are raising interest in Islamic Fintech in recent years, the area is still relatively undiscovered, and empirical proofing is required in many issues. Last of all, fintech has enormous potential for Islamic banking. However, this rapidly growing fire requires a large and continuous flow of wood for the long run to continue. If human resources, academic research, a healthy ecosystem, governmental vision, and regulatory bases are provided, Islamic fintech will be the future of Islamic finance.

b.    Digital Currencies, Blockchain, and Other DLT Technologies in Islamic Finance

Blockchain technology has gained significant attention in the past decade by overcoming a critical technicality in human nature. Al Ghazali argues that human nature has several components: varying between virtuous and evil [24]. In the current environment of the modern world, evil components are fed more, and humankind has become less trustable [25]. Ethical gaps in the 2008 sub-prime crisis exposed this fact sardonically, and blockchain technology, specifically bitcoin, was born from this painful environment.

Blockchain technology is based on a decentralized ledger for recording activities; thus, none of the included parties are trusted more than the others [7]. Each transaction is recorded in a block of information. All blocks are added and written in all users' data packs simultaneously, making it nearly impossible to cheat on the system [26]. Therefore, blockchain-based techniques can and should be used in Islamic finance transactions, increasing global trust in Islamic finance at this critical stage [27]. As mentioned, Islam encourages transparency and honesty in all dealings, and blockchain is a valuable way of ensuring it.

Blockchain is so flexible that it could be merged with almost any other technology such as AI, machine learning, rule-based and statistical analysis [28,29], image processing [30], machine translation [31], chat-boxes [32], artificial neural networking [33,34], etc. In the Islamic finance world, such as conventional, new technologies are adopted by younger fintechs and start-ups, and later they are merged or acquired by larger institutions or banks. This is often in acquiring a service or an internal system, such as auditing or organizing the operations. For example, using blockchain, specifically smart contracts in the agent auditing and monitoring systems with the required software, would help ensure the transparency of all parties in Islamic banks, including shariah boards, beneficiaries, regulators, and management [35,36].

Other studies focus on using blockchain-based systems for different areas of Islamic finance, such as crowdfunding, microfinance, and Islamic philanthropy management [37]. Muneeza [38] researched the usability of blockchain in Islamic crowdfunding transactions. Crowdfunding is a sensitive area with razor-thin margins, and reducing transaction costs and frictions is vital. Blockchain and cryptocurrencies eliminate cross-border costs and allow instant transactions, thus taking a life-saving role in crowdfunding [39]. Zhu and Zhou [40] also focused on cost reduction by using blockchain-based technologies in fundraising for several purposes. Transparency and cost reduction increase the trust in the fundraising institution and eliminate country limitations by global usage of digital currencies. In addition, by eliminating a central body of these businesses, global reachability is at its current top level with zero cross-border costs, transaction costs, time delays, and minimum legal limitations [41,42].

Blockchain is also used to detect fraud in crowdfunding and fundraising businesses by constantly auditing the ledger [43]. This face of blockchain technologies is helpful for governments and other regulatory institutions by easing their operations. Frauds in other banking transactions and terrorism funding can be detected using digital currencies, specifically central bank digital currencies (CBDCs). More of CBDCs discussions will be in upcoming paragraphs.

Cryptocurrencies, CBDCs, and specifically bitcoin are essential in blockchain research. Moreover, these areas have a critical role in Islamic finance as well. Besides easing banking operations and overcoming cross-border frictions, digital currencies are helping Islamic finance to apply Maqasid al-Shariah around the globe significantly. One of its main tasks is cost reduction and increasing reachability and inclusion, which is essential in protecting wealth (*mal*) in Maqasid [44]. However, a critical point worth mentioning here is the choice of cryptocurrency, as some have a hot debate over their Islamic permissibility [45].

However, some studies pay attention to one of the main problems of all digital currencies, double spending. Double spending occurs when the same entity spends twice the same unit of money. In centralized payment systems, the center holds a logbook for the transactions. Hence, each spending is recorded. However, in a decentralized system, there is no center for a ledger. To overcome double spending, any transaction must be approved by all joining peers and then recorded in the ledger [46].

Othman [47] mentioned that if not applied by banks, cryptocurrencies could challenge banks' deposit account volume. This is because cryptocurrency investments are pulled from banks' deposit accounts and therefore cause a shift in investment trends [48]. Othman [47] suggests that banks should offer portfolio diversification options in cryptocurrency form or directly implement blockchain-based currencies in their system to overcome this challenge.

However, in this case, experts share their concerns over the unknown risks of cryptocurrencies as they are very young and premature and could cause catastrophic results for banks [49].

Besides their usefulness and convenience, cryptocurrencies, especially Bitcoin, receive less attention from the scholarly community regarding their unknowns and excessive risks [50–52]. Cryptocurrencies' research output focuses on technical details or their risky aspects. Carrick [53] argues that cryptocurrencies must be a supplemental monetary system besides fiat money, but not as the new money that replaces the current one. However, a heterogeneous market structure would bring its own risks, as Abedifar et al. [54] discussed.

From the Islamic community, the risks and unknowns of cryptocurrencies are partially understood, which could be understood from the fatwa of several Shariah scholars of Muftis. Influential Shariah scholars from countries such as Egypt or Indonesia have declared that all cryptocurrencies are haram. However, Rabbani et al. [7] argue that not every Islamic scholar understands these products' underlying structure. The number of cryptocurrencies is growing daily, moving around 20,500 bands [55]. However, not each of these could be classified as currency, but some have different structures. Therefore, the name "crypto assets" would be more inclusive than cryptocurrencies [7]. Although currencies such as Bitcoin have several Shariah issues and cannot be used in Islamic transactions, there are certainly many other options in the crypto asset world for this purpose [45,56].

From a Shariah perspective, mainstream cryptocurrencies have a critical Gharar element. The high volatility, unknown real value, and improper store-of-value methods bring up this element [27]. The Arabic word Gharar means excessive uncertainty and risk and is prohibited in Islamic transactions and contracts [57]. Another study focuses on the lack of intrinsic value and supervision of a central bank over the cryptocurrencies; therefore, they would be against Shariah from a social justice perspective [58].

However, some literature argues the opposite, that cryptocurrencies, even the mainstream ones, could be Shariah permissible, and none of the arguments related to their impermissibility does not have a strong basis. Oziev [59] mentions these points and argues that the record jumps of bitcoin are not associated with its Gharar technicality but are related to its fast-growing popularity and era-opening innovation. Moreover, from a Maslahah perspective, these coins could develop Islamic society in several dimensions, increase the overall wealth, and therefore serve the Maqasid. Indeed, the risk received in cryptocurrency investment could be minimized by joining partnerships by individuals, which would apply the structure of Musharakah [36,56].

While discussing cryptocurrencies, whether they are Shariah permissible or not, the concept of a digital coin issued by a central bank with an actual intrinsic value and underlying asset seems to solve most of the discussed problems. In a direct model, the central bank digital currency (CBDC) [39] concept allows users to start their accounts directly from the central bank, not with the banks or other institutions [60]. Such a system requires massive investment and a distributed ledger protocol. Therefore, the running costs of a consensus mechanism cause DLT-based infrastructures to have a high investment required, making it unattractive for direct claimed CBDC structures. Except for small regions where the daily number of transactions is manageable by a DLT-based central bank infrastructure, conventional central bank infrastructure is preferable for directly claimed CBDC designs. Several existing CBDC experiments in different countries also give similar results on this technicality [61]. Moreover, unlike a decentralized private coin, a consensus mechanism would get under pressure quickly with a service denial type of attack, making it risky to use as an infrastructure of a whole economy [62].

c.    Digitalization of Islamic Non-bank Financial Institutions

Digitalization of the financial world brings excellent opportunities for institutions from every layer. Digital opportunities of the 21st century would have been perceived as magic a few centuries ago. In his famous book, first published in 1967, Clarke [63] formulated three laws about technology, where the third law is the most famous one; "Any

sufficiently advanced technology is indistinguishable from magic." Today we can feel this fact dramatically in the finance world.

The advancements that came with fintech development are also consistent with the main pillars of Islam. For example, transparency is one of the most exciting features of blockchain systems, which is also one of the top concerns of Islamic business ethics [17,18]. Hence Islamic finance and fintech have grown enormously after the subprime crisis. Customers lost their trust in conventional financial systems, where tranches of bonds and other securities are mixed and marketed as AAAs [64]. Shariah has clear lines about the transparency and ethical ruling of businesses, and therefore Islamic banks are required to keep their businesses in line to protect their credibility. Literature also indicates that the increasing competition between conventional and Islamic finances helps the financial world become more trustable [19]. This role is expected to be taken by Islamic fintech primarily because of its potential and growth. Islamic fintech is expected to earn over 150 million new customers in the next three years, and businesses and researchers should work hard to protect this potential [20].

Institutions would also benefit from fintech and digitalization to keep their internal structures trustable. Shariah supervisory boards, internal audit systems, and regulatory frameworks could all be organized by blockchain-based solutions. Moreover, today's smart contract-based auditing systems can work completely automated, reducing the required labor for any institution. Although Shariah seems very flexible about decision-making in new products, the methodology is prominent and robust, making it possible to create an AI-based system that can decide without humans [65].

Many scholars perceive smart contracts as the future of business law [66], not only for Shariah's decision-making, but also for almost any business agreement that can be based on blockchain smart contracts, thus skyrocketing its potential. Although smart contracts are still not legally accepted, legal authorities agree on their potential and usability in the future [7]. Smart contracting brings speed and ease to asset ownership transactions while minimizing transaction costs by over 95% [67]. With all their potential, scholars agree that smart contracts can overtake all the burden of legal systems and regulatory works by automizing them [68].

Another main benefit of digitalizing financial institutions is the cost reduction in time, labor, and money. As mentioned, some Islamic financial institutions have razor-thin margins, and cost reduction is their survival strategy. Especially international transactions are very challenging, as they bring in new costs, such as cross-border payments [69]. Especially blockchain and crypto assets bring groundbreaking solutions for these institutions, where a seamless global hub of money can easily be created.

Moreover, as mentioned in the holy Qur'an (9:60), Zakat has a limited area of respending to 8 main groups, which are the needy, the impoverished, Zakat charities, reverts, and the friends of the Muslim community, the enslaved, the indebted, stranded travelers, and those who fight on behalf of Allah. Although these categories are important pressure points for societal empowerment, the financial use of Islamic philanthropy is limited in terms of financial area [70]. Today, there are many initiatives for blockchain-based fund management and zakat collection. Indeed, besides the zakat collection, another main challenge is to distribute the collected Zakat accurately. Digital zakat distribution and classification systems are produced in Malaysia and Indonesia to correctly categorize people in need and determine the required fund percentages for each category [71].

As competition is getting more intense, the speed of the transactions has become customers' primary source of preference. Pernot [72] explains that the average waiting time of customers per click is no more than a few seconds, meaning that customers would not prefer a system that keeps processing transactions for more than a few seconds. Especially Generation Z has very high expectations of speed, ease, and smooth interface, which makes it very difficult for institutions to update themselves while staying competitive [10]. Without an efficient use of digital systems, blockchain, and several other concepts such

as AI, machine learning, big data, IoT, etc., any institution would immediately fade away from the markets.

Indeed, the literature discusses the risks coming with digitalization. The first and most critical risk for a highly digitalized financial institution is that it will be exposed at every level of its structure. This risk could be rapid customer shifts, bank runs, system crashes, or cyber-attacks [73]. Moreover, fintech is still very young, and trends shift from one extreme to another quickly, making it challenging to take a strong base for institutions [17]. This is the main reason large institutions do not base their systems solely on fintech but establish partnerships with young fintech institutions or acquire them to utilize the services. With this strategy, the institution's main business will stay safe even if the fintech trends shift rapidly.

## 4. Concluding Remarks

This study has completed a systematic review of Islamic fintech and digitalization literature, which is highly required because of the tremendous speed of developments in this field. Although the area would be more extensive, the study has focused on Islamic banking digitalization, blockchain and other crypto assets, and Islamic non-bank financial institutions' digitalization.

Overall, research shows that fintech has a deep relationship with the primary doctrines of Islam and Islamic finance, which supports fintech investment by Islamic financial institutions. Transparency, social benefit-focused investment, protecting the wealth of the people and the environment, protecting the human factor, and minimizing the friction in terms of money, labor and time are some of the common goals for both fintech and Islamic finance.

Moreover, blockchain and other DLT-based technologies are now well-known within Islamic finance. These technologies could serve Islamic financial institutions by helping to apply Maqasid rulings within their operations. The results of this literature review clear out the relationship between these two areas, encouraging the fintech investment by Islamic financial institutions to reach their goals more effectively.

Another focal point of the research focuses on the opportunities for Islamic fintech to reach SDGs by increasing efficiency, financial inclusion, and by reducing costs. Moreover, as discussed in the paper, fintech investment reduces overall energy usage and better protects the environment. The recent decision of Ethereum to leave the proof-of-work concept and start using proof-of-stake is an appropriate example of this development because it reduces the required energy usage of mining by a significant amount.

Blockchain technologies are flexible enough to be merged with any other fintech tool, therefore bringing a new perspective into the fintech world. Islamic finance also takes its pie from blockchain disruption; both banks and non-bank institutions benefit from it deeply. Islamic finance research focuses on blockchain and other DLT-based technologies more than ever, and the results of this research show that the efficiency and speed that Islamic finance needs could be achieved by these technologies relatively quicker than conventional fintech.

Islamic fintech has great potential mainly because of the overlapping norms of Shariah and fintech, making it easier to implement technological advancement into Islamic finance. Moreover, the trust shift to Islamic finance could be merged with the opportunities of fintech and increase the potential of Islamic fintech even more. Fintech and blockchain bring transparency and convenience into operations, which is one of the main goals of Shariah. A genuinely Islamic business should be transparent and efficient and protect people's wealth. Keeping a business Shariah compliant is more difficult while keeping it competitive in today's harsh markets.

Islamic fintech, combined with smart contracts, will also ease the regulatory burden by automating transactions at any level. Many scholars perceive blockchain-based contracting combined with crypto assets as the future of finance. Today, most Muslim countries have great potential in their resources and populations. They should invest more time and labor into digitalization and fintech research to use their potential more efficiently.

**Author Contributions:** Conceptualization, A.F.A. & I.M.U.; methodology, A.F.A. & I.M.U.; software, A.F.A. & I.M.U.; validation, A.F.A. & I.M.U.; formal analysis, A.F.A. & I.M.U.; investigation, A.F.A. & I.M.U.; resources, A.F.A. & I.M.U.; data curation, A.F.A. & I.M.U.; writing—original draft preparation, A.F.A. & I.M.U.; writing—review and editing, A.F.A. & I.M.U.; visualization, A.F.A. & I.M.U.; supervision, A.F.A. & I.M.U.; project administration, A.F.A. & I.M.U.; funding acquisition, A.F.A. & I.M.U. All authors have read and agreed to the published version of the manuscript.

**Funding:** This research activity/output was supported under the HBKU CIS' Research Clusters Grant.

**Institutional Review Board Statement:** Not applicable.

**Informed Consent Statement:** Not applicable.

**Data Availability Statement:** Not applicable.

**Conflicts of Interest:** The authors declare no conflict of interest.

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
