# Peer review of "Fintech, Digitalization, and Blockchain in Islamic Finance: Retrospective Investigation"

_fintech, doi:10.3390/fintech1040029_

Round 1

Reviewer 1 Report

The article does not bring anything new. Just the presentation of already known information. 

It is similar like report, not research article. No literature review and methodology.

 Introduction: Overall, there is a lack of argumentation on why it is important from a theoretical perspective.

There is no clear description and understanding why these different concepts are being discussed. It is important for the authors to understand that the essence of a literature review is not only to summarize text but also evaluate and clarify what previous research have found about the phenomenon you are studying. There is also no clear connection how the different part of the literature are related to each other. The author/authors also need to be careful that proper references are done. Fourteen self citation is not acceptable for any reputed journal. 

Author Response

Dear respected referee,

Thank you for your detailed review. You may refer to the attached file for the reply.

Regards.

Reviewer 2 Report

This paper aims to review academic publications (journal articles) to understand the trends related to Islamic banks’ digitalization, Blockchain & Crypto Assets research, and Islamic non-bank financial institutions’ digitalization.

The article will benefit greatly if it is re-arranged to follow the standard presentation:

1. Introduction. This section gives the motivation and objective of study,

2. Data and Methodology. Here the database of academic publications obtained, total number of articles, number of articles for each category. Figures 1 and 2 will be here.

3. Results. Discussion on results divided into 3 categories (i) Islamic banks’ digitalization, (ii) Blockchain & Crypto Assets research, (iii) Islamic non-bank financial institutions’ digitalization. It is recommended that Tables (listing the academic publications by authors, years, data base) be used in this section to clearly justify the specific criteria chosen by authors to describe the academic publications that fit each category.

4. Conclusion / Concluding Remarks. Include future research for this study.

Author Response

Dear respected referee,

Thank you for your review. You may refer to the attached file for the reply.

Regards.

Round 2

Reviewer 1 Report

The paper is still not in publishable shape. It is more like a thesis. Literature review is after method, that define the naive approach of authors. 

Reviewer 2 Report

The article has been improved accordingly. It is suggested that authors send paper for English editing before publication.